# Comparative Efficiency of Native Insect Pollinators in Reproductive Performance of *Medicago sativa* L. in Pakistan

**DOI:** 10.3390/insects12111029

**Published:** 2021-11-15

**Authors:** Abdur Rauf, Shafqat Saeed, Mudssar Ali, Muhammad Hammad Nadeem Tahir

**Affiliations:** 1Institute of Plant Protection, Faculty of Agriculture and Environmental Sciences, MNS University of Agriculture, Multan 60000, Pakistan; mudssar.ali@mnsuam.edu.pk; 2Institute of Plant Breeding and Biotechnology, Faculty of Agriculture and Environmental Sciences, MNS University of Agriculture, Multan 60000, Pakistan; hammad.nadeem@mnsuam.edu.pk

**Keywords:** solitary bees, honeybees, syrphid flies, visitation rate, single visit efficacy

## Abstract

**Simple Summary:**

Lucerne (*Medicago sativa* L.) is a cross-pollinated crop and pollination does not usually take place until the sexual column is released. The discharge of the sexual column is referred to as ‘tripping’ the flower. In this study, several bee species were investigated to facilitate the pollination of lucerne crop with a purpose to examine the pollination efficiencies of native insect pollinators that lead to a seed set of alfalfa. We designed a two-year field trial to investigate the abundance and diversity of insect pollinators along with foraging behavior in terms of stay time, visitation rate, pollen harvest and tripping efficiency. Moreover, the single-visit efficiency in terms of number of seeds per pod, germination and seed yield was evaluated. Ten major pollinators (five solitary bees, three honeybees and two flies) were tested for their pollination efficiency as effective pollinators. Findings show the solitary bees (*Megachile cephalotes*, *Megachile hera*, *Amegilla* sp.) can be recommended as an effective pollinator for this crop, thus increasing options available to seed growers. For lucerne seed development, more consideration should be paid to the variety of native wild bees, conservation strategies and foraging requirements in establishing a diverse pollination system rather than a single bee species.

**Abstract:**

Lucerne (*Medicago sativa* L.) is a cross-pollinated crop and requires entomophilous pollination for tripping of flowers and subsequent pod and seed set. To discover the best pollinators for lucerne seed production, a two-year field trial was carried out at the research farm of MNS University of Agriculture, Multan, Pakistan. Abundance and diversity of insect pollinators along with the foraging behavior were recorded in terms of tripping efficiency, stay time, visitation rate and pollen harvest. The single-visit efficiency of abundant insect pollinators was also evaluated in terms of number of seeds and seed weight per raceme along with germination percentage. Ten most abundant floral visitors (five solitary bee species, three honeybee species and two syrphid fly species) were tested for their pollination efficiency. Honeybees were most abundant in both the years followed by the solitary bees and syrphid flies. Single-visit efficacy in terms of number of pods per raceme, number of seeds per raceme, 1000 seed weight and germination percentage revealed *Megachile cephalotes* as the most efficient insect pollinator followed by *Megachile hera* and *Amegilla* sp. Future studies should investigate the biology and ecology of these bee species with special emphasis on their nesting behavior and seasonality.

## 1. Introduction

Insect pollination is essential for the sustainability of both agricultural and natural ecosystems since 87% of flowering plants and nearly 35% of crops worldwide are dependent on insect and other pollinators [1,2]. Bees have been reported as the most efficient among the insect pollinators due to their distinctive characteristics [2,3,4]. Both managed and wild bees provide pollination services to the vast majority of fruits, vegetables, forage crops (alfalfa and clovers), oil-producing crops [5] and wild flowering plants [6].

Lucerne, also known as alfalfa (*Medicago sativa* L.), is a cross-pollinated crop. Because of its high nutritional content, adaptability, good quality characteristics and herbage yield, it is one of the most valuable source as fodder grown in more than 80 countries [7,8,9,10,11,12,13]. Pollination is a key limitation in alfalfa seed production.

The lucerne flower has a standard petal on which bees often land, as well as two smaller wing petals on each side. These keel petals exert significant pressure on the female sexual column. The flower of alfalfa is pollinated unless the keel is forced to open i.e., the release of sexual column. The discharge of the sexual column is referred to as ‘tripping’ the flower [14]. Lucerne flowers need visiting bees to trip the sexual column, allowing for eventual pod and seed set [15,16,17]. Due to their unique floral structure and pollination requirements, a wide array of insect pollinators have been reported to visit lucerne flowers, including bees e.g., *Apis cerana*, *A. florea*, *A. dorsata*, *Xylocopa* sp.; *Megachile lanata*, *M. rotundata*, *M. bicolor*; *M. abluta*, *Nomia melanderi*, *Andrena lebedevi* and flies, e.g., *Eristalinus megacephalus*, *E. obliquus* [18,19,20,21,22,23,24,25,26]. However, honeybees cannot pollinate lucerne flowers effectively [27,28]. This inefficiency of honeybees in lucerne pollination is because of some adaptive behavioral characters, its side feeding habit for only nectar foraging and lucerne pollen lacking the essential amino acid, isoleucine [29,30,31,32,33].

Reproductive success in lucerne is influenced significantly by pollinator abundance, their foraging behavior and pollination characteristics (through the provision of pollen and nectar) [22,26,34,35,36,37]. Single-visit seed set is an important parameter for assessing the effectiveness of insect pollinators [38,39,40,41,42]. Some behavioral and morphological parameters such as stay time or visit duration and pollen load affect yield attributes [29]. The efficiency of bees can be assessed in terms of seed set after their single visits [43,44]. Lucerne seed production can be obtained through commercially available bees (*Megachile rotundata* F. and *Nomia melanderi* Cockerell) or through available native bees around the world [15]. Only a few studies have reported biodiversity of native pollinators and their role in reproductive success (seed set) of lucerne [15,26,45], while no previous study has reported the single-visit seed set efficiency of native insect pollinators for lucerne seed production.

The purpose of this study was to assess the biodiversity of native insect pollinators (i.e., solitary bees, honeybees and flies) along with comparing their pollination efficiency in order to find the best native pollinators for lucerne seed production.

## 2. Materials and Methods

### 2.1. Study Site and Experiment Design

This study was conducted at the experimental farm of MNS University of Agriculture, Multan (longitude 71° 26′ 35.5″ E and latitude 30° 08′ 50.5″ N), Punjab, Pakistan. The climate of Multan is sub-tropical (arid and semi-arid) with hot summers and cold winters with low rainfall [46]. The maximum and minimum mean temperatures in summer are 42 °C and 29 °C, respectively, while these are 21 °C and 4.5 °C in winter, respectively. The average annual rainfall is around 186 mm, with the majority of occurrence during the monsoon season from July to September [47,48]. The soil of the area is classified as sandy loam that is mildly alkaline and well-drained [49]. The lucerne cultivar ‘Sargodha 2000’ was sown on an area of one acre during November in both the years (2019 and 2020). The row-to-row spacing was maintained at 30–45 cm. All the standard cultural practices were used in the experimental plot except the use of any pesticides [50].

### 2.2. Study Plant Species

Lucerne is widely grown in Pakistan as a fodder crop in different districts of Punjab and Sindh provinces. It is a perennial outcrossed open-pollinated legume pollinated by insects. Flowers are bisexual, with four petals and ten stamens arranged into racemes, and a plant can support several stems, each with multiple racemes and flowers [16,51]. Individual flowers remain open for a week if no pollination occurs and multi-seeded pods will mature six weeks after pollination. The peak bloom occurs between April and June in South Punjab, Pakistan [52].

### 2.3. Diurnal Abundance of Insect Pollinators

To measure the abundance of insect pollinators, three quadrats were randomly selected by throwing one square meter wooden frame during the blossom period of alfalfa. Counts were taken once a week at three-hour intervals throughout the day, starting from 8:00 am to 17:00 pm across the whole flowering season (April–June). During each time interval, a quadrat (one square meter) was observed for five-minute and all the insect visits were counted [53]. All the bee specimens were first morphotyped and then identified on wings during data recording. We also observed many other flower visitors, which did not come under systematic observation. Insect specimens were identified to the lowest taxonomic level by the experts (see Acknowledgments). All the voucher specimens were preserved and submitted to the Agricultural Museum of the Faculty of Agriculture and Environmental Sciences, MNS- University of Agriculture, Multan, Pakistan.

### 2.4. Foraging Behavior of Insect Pollinators

The tripping efficiency of abundant insect pollinators was observed in terms of the following parameters: the number of virgin flowers tripped per raceme, flowers visited but not tripped per raceme, and visiting already tripped flowers per raceme [15,24]. The number of flowers visited by a pollinator whose stigmas and anthers released from the keel (referred to as the tripping). The weekly data were recorded throughout the flowering season at 8:00, 11:00, 14:00 and 17:00 h.

Pollen harvest was carried out early in the morning. The buds were caged one day before their opening with a nylon mesh bag. These fully opened flowers were uncaged and visiting individuals were captured after a single visit. The pollen grains were counted in the laboratory under a stereo-zoom microscope by hemocytometric method [54].

Moreover, visitation rate (numbers of raceme visited/60 s) and stay time (the time spent per single raceme) of insect pollinators was also recorded. Weekly observations were carried out at 8:00, 11:00, 14:00 and 17:00 h, since different insects had different diurnal dynamics [55,56,57,58].

### 2.5. Single-Visit Seed Set Efficiency

For measuring single-visit seed set efficiency, some unopened floral racemes were caged with nylon mesh bags (15 × 7 cm) before their opening. Upon opening of flowers, the mesh bags were removed and an individual of a specific pollinator species was allowed to land on the raceme and to visit the flowers. The racemes were re-caged after the insect visit and the un-tripped flowers were removed. Ten single visits were made for each pollinator species. Some flowers were also manually pollinated/tripped by using experimental hand pollination. The resultant reproductive success was recorded in terms of number of seeds per raceme, seed weight per raceme and germination percentage. Fifty open-pollinated (unrestricted insect visitation) flowers and 50 caged (no insect visitation) flowers were also maintained for the comparison. The resultant seeds were subjected to germination test by placing all seeds on moist filter paper in a Petri dish at room temperature, and germination was checked four days later [59].

### 2.6. Data Analysis

Data regarding pollination effectiveness of different species in terms of tripping, pollen harvest, foraging behavior (in terms of visitation rate and stay time) and single visit efficiency of abundant insect pollinator (in terms of number of pods per raceme, number of seeds set per raceme, 1000 seed weight) and germination (%) were initially tested for normality as the basic assumption of ANOVA is that error of the model should be distributed normally. By using QQ-plot and normality test, we did not find this assumption to be met for our data. So, we used nonparametric methods to test the mean equality. These methods are based upon ranking of the mean. Thus, we used Kruskal–Wallis test for analysis the data followed by Conover-Iman post-hoc test as corresponding Kruskal–Wallis null hypothesis were rejected. Data were analyzed by using computer software XLSTAT (XLSTAT. 2014 version: 5.03).

## 3. Results

### 3.1. Pollinator Community/Floral Visitor Census

In this study, honeybees were most abundant followed by solitary bees and flies in both years. Apidae was the dominant family with five species, followed by two species of Megachilidae and a single species from the Halictidae. The pollinator community of lucerne was composed of 16 bee (Hymenoptera) species and six true fly (Diptera) species. A smaller proportion of these species was found during our systematic observations, i.e., five solitary bee species, three honeybees and two true fly species. Among the bees *Megachile bicolor*, *Megachile lanata*, *Pseudapis* sp., *Xylocopa fenestrata*, *Xylocopa basalis*, *Halictus* sp., *Ceratina* (Pithitis) *smaragdula* and *Andrena* sp. whereas, among flies *Eristalinus laetus*, *E. megacephalus*, *Chrysomya rufifacies* and *Musca domestica* were rarely seen.

The *Amegilla* sp., *Eucera* sp., *Nomia* (Hoplonomia) sp., *Megachile hera* and *M. cephalotes* were the most abundant visitors during both the studied years. Among solitary bees *Eucera* sp. and *M. cephalotes* were the most frequent floral visitors with totals of 215 and 206 individuals, respectively for both years. The average visitation frequency was also highest among all the observed bee species: (0.029, 0.042) and (0.022, 0.047) individuals per square meter per minute for the year 2019 and 2020, respectively (Table 1).

*Apis florea* was most abundant (472) among honeybees while *Eristalinus aeneus* was the most frequent floral visitor (481) among the flies during both years studied. Diurnal dynamic patterns revealed the peak abundance of all the insect pollinator groups was attained at 11:00 am to 14:00 pm followed by a gradual decline until 17:00 pm during both years (Figure 1).

### 3.2. Pollination Effectiveness

There was a difference among the three pollinator groups in terms of their tripping efficiency (chi-square = 32.69, DF = 39, *p* = 0.0000) and also visiting already tripped flowers (chi-square = 32.09, DF = 39, *p* = 0.0000). Tripping efficiency of solitary bees was higher than the honeybees and flies. Flies and honeybees preferred to visit the already tripped or open flowers during their foraging.

Pollinator species varied with respect to single visit pollen harvest from the lucerne flowers (chi-square = 74.18, DF = 9, *p* = 0.0000). The pollen harvest was found to be highest for two *Megachile* species (*M. cephalotes* and *M. hera*) followed by three honeybee species (*A. mellifera*, *A. dorsata* and *A. florea*) while it was lowest for syrphid fly (*E. aeneus*) (Table 2).

The probabilities of tripping flowers change over time: more flowers were tripped by pollinators on an inflorescences at 11:00 am (33.63 ± 4.45) and by 8:00 am (33.87 ± 3.73) while no difference was observed between 14:00 pm and 17:00 pm. Moreover, pollinator visits to already tripped flowers were higher at all times of the day except at 17:00 am as shown in Table 3.

### 3.3. Foraging Behavior

The visitation rate varied among the tested pollinator species (chi-square = 22.88, DF = 9, *p* = 0.01). *Eucera* sp. visited the maximum racemes per 60 s (13.75 ± 1.38) followed by *Amegilla* sp. (12.40 ± 1.38) and *M. cephalotes* (12.32 ± 1.41).

There was a difference between the pollinators species in terms of their stay time on the raceme (chi-square = 39.39, DF = 9, *p* = 0.00). *Eucera* sp. and *Nomia* (Hoplonomia) sp. spent maximum time on raceme (17.66 ± 4.31 and 13.70 ± 20.3) followed by *A. florea* (12.02 ± 1.92) and *M. hera* (10.40 ± 1.59) (Table 4).

### 3.4. Single Visit Effectiveness of Pollinators

There was a difference among the pollinator species in terms of number of pods produced per raceme (chi-square = 74.22, DF = 23, *p* = 0.0001), number of seeds per raceme (chi-square = 78.62, DF = 23, *p* = 0.0001), 1000 seed weight (chi-square = 49.60, DF = 11, *p* = 0.0001) and germination (chi-square = 47.08, DF = 11, *p* = 0.0001) among the tested pollinator species in a single visit. Single visit efficacy proved *M. cephalotes* as the most efficient insect pollinator followed by *M. hera* and *Amegilla* sp. while the flies (*E. arvorum* and *E. aeneus*) were the least efficient insect pollinators. Open pollination treatment resulted in highest number of pods per raceme, number of seeds per raceme, 1000 seed weight and germination. Hand tripping also proved less effective than the single visits of three most efficient bee species. No pod development was observed in pollination exclusion treatments (Table 5).

## 4. Discussion

Pollination efficiencies of leguminous crops (alfalfa and clover) differed greatly among pollinator species. In our study, honeybees were the highest in proportional abundance (45%) in the first year, while honeybees and solitary bees (35%) had the similar proportional abundance in the second year. However, in both years, syrphid flies were the least abundant. Some previous studies have also reported higher abundance (85–98%) of solitary bees visiting lucerne flowers [24,26,30,45,60]. Moreover, from the same region, the higher abundance of honeybees and solitary bees has also been reported from other crops, i.e., canola [58], pumpkin [61]; luffa gourd [62] and radish [63]. In our study, during both years, bees from the family Megachilidae were more abundant (44–47%) as compared to the other bee families. Worldwide, higher abundance of *Megachile* bees has been linked with the enhanced alfalfa seed production due to their higher pollination efficiency [5,16,22,30,64].

Pollinator species varied in terms of tripping lucerne flowers and it was higher for solitary bees than the honeybees and flies. Among solitary bees, two *Megachile* sp. (*M. cephalotes* and *M. hera*) and *Nomia* (Hoplonomia) sp. were superior in tripping as compared to other most-abundant bee species. These tripping rates varied among pollinator species [24]. Some previous studies have also reported the better tripping efficiency of *Nomia melanderi* and other *Megachile* species (*M. rotundata* and *M. abluta*) on alfalfa flowers [15,45]. Social characters of *Megachile* sp. to collect pollen from the lucerne, its maximum foraging period coinciding with the lucerne blooming, also enable it to trip the lucerne flowers more efficiently [65,66]. Since most of the female solitary bees are pollen foragers, they exploit flower by ensuring tripping of *Medicago* flowers [31]. Moreover, tripping efficiency of *Nomia* sp. is due to its buzz pollination character, production of several generations in a year, high abundance and foraging time synchrony with lucerne blooming [25]. Tripping varied with time of day; our results are in line with the previous studies illustrating that visits of bees peaked from 10:00 am to 14:00 pm, and this period coincided with the complete release of the pollen [32]. Moreover, all the pollinators were at their peak abundance at 11:00 am to 14:00 pm during both years. This is attributed to the pollinators’ activity, which is predicted by the time of anthesis, the time when a large amount of nectar and pollen resources are available [33].

In our study, both managed and wild honeybees were least effective (4–10%) in terms of tripping the flowers. Some previous studies have also reported the inefficiency of managed honeybees in tripping the alfalfa flowers due to different reasons. First, honeybee workers quickly learn to access the nectar without tripping the flower due to their side feeding habit that minimizes the chances of tripping lucerne flowers [31,67]. Second, lucerne pollen is lacking the essential amino acid (isoleucine), which means honeybees suffer protein stress when only pollinating lucerne [30]. Third, some adaptive behavioral characters also lead honeybees towards poor tripping, resulting in 1% tripping in open field conditions [29,68,69] and only 22% under caged experiments [15]. Moreover, tripping inefficiency (5% of the visited flowers) of wild honeybees (*A. dorsata*, *A*. *florea*) has also been reported [24,70].

Similarly, syrphid flies (*E. aeneus* and *E. arvorum*) also proved ineffective in terms of tripping lucerne flowers. Previously, tripping inefficiency of syrphid fly (*Eristalinus tenax*) and other flies (*Calliphora vicina* and *Lucilia sericata*) has also been reported from *Medicago* pollination [71]. Syrphid flies feed mainly on nectar and often return to feed on previously visited flowers, despite being caged and restricted in space or resources [72,73,74].

In the current study, solitary bees harvested more pollen on a single visit than those of honeybees and flies. The two *Megachile* species (*M. cephalotes* and *M. hera*) harvested the highest number of pollen grains followed by *A. mellifera* and *A. dorsata*. This pollen harvest indicates a passive pollen load by the bees on a single visit. The pollen-foraging individuals can be more effective pollinators than the mere nectar foragers [75,76] and pollen foraging leads to greater tripping of alfalfa flowers [77,78]. In our study, *Nomia* sp. and *Eucera* sp. were less efficient in terms of harvesting pollen grains although their tripping efficiency was equivalent to the other high pollen harvesting species. Contrarily, a previous study reported high pollen harvest (2100) for *N. melanderi* after their 15 visits in *M. sativa* [15].

Solitary bees proved best in terms of single visit pod set per raceme and seed setting as compared to honeybees and flies. Among the solitary bees, *M. cephalotes* resulted in maximum pods and seed set followed by *M. hera* and *Amegilla sp*. Probabilities of pod and seed setting are higher for the insect pollinators with higher tripping rates [15]. Pod setting done by *Nomia* (Hoplonomia) sp. and *M. hera* was 66 and 62%, respectively. Previously, a single study reported effectiveness of *M. rotundata* and *N. melanderi*) as most efficient in terms of single-visit pod set efficiency in the range of 40–57% [15]. *Megachile rotundata* and *N. melanderi* are the intensively managed bees on commercial level for lucerne crop [20]. Moreover, from South Punjab region, the effectiveness of other solitary bees in terms of single-visit seed/fruit set have been reported for other crops, i.e., canola [58], bitter gourd [56], pumpkin [61], luffa gourd [62] and falsa [79].

The effect of solitary pollinating insects was remarkable and confirms the preponderant role of solitary bee pollinators on seed set of *Medicago* flowers whereas honeybees were confirmed to be poor pollinators of this crop, which collect pollen from *Medicago* under certain circumstances. Pollen-gathering honeybees are usually better pollinators than nectar-gathering honeybees, which often forage by ‘side-working’ the flowers. During those visits, the bee does not make contact with the stigmas. Solitary bees, on the other hand, usually work the flowers from the tip, irrespective of foraging activity, almost always contacting the anthers and stigmas [80,81,82,83]. Moreover, flies (*E. aeneus* and *E. arvorum*) were also ineffective in their single-visit seed set in both the years. Seed setting of flies *E. tenax* (Syrphidae), *C. vicina* and *L. sericata* (Calliphoridae) were lower because of insufficient pollen transfer and pollen adhering to their bodies [71].

In open pollination, seed set was 85% higher as compared to the self-pollination (0%). Self-pollination is not fruitful in the case of lucerne seed formation due to the abortion of seeds [15]. Previously, open pollination along with solitary bees produced significantly more seeds as compared to honeybees and pollination-exclusion flowers [18,26,84]. Moreover, open pollination also produced 87% higher seed set than the hand-tripped flowers. Previous hand-pollination studies in alfalfa have shown that hand-tripped flowers generally produce pods with one to two seeds per pod [26,71,85,86,87].

## 5. Conclusions

In conclusion, the solitary bees (*M. cephalotes*, *M. hera*, *Amegilla* sp.) proved to be the most efficient pollinators for lucerne seed production. Conservation of these most-efficient bees (through provisioning of foraging and nesting resources) can lead to sustainable alfalfa seed production. Future studies should focus on exploring the population threshold of these bee species for better pollination and seed production in lucerne.

## Figures and Tables

**Figure 1 insects-12-01029-f001:**
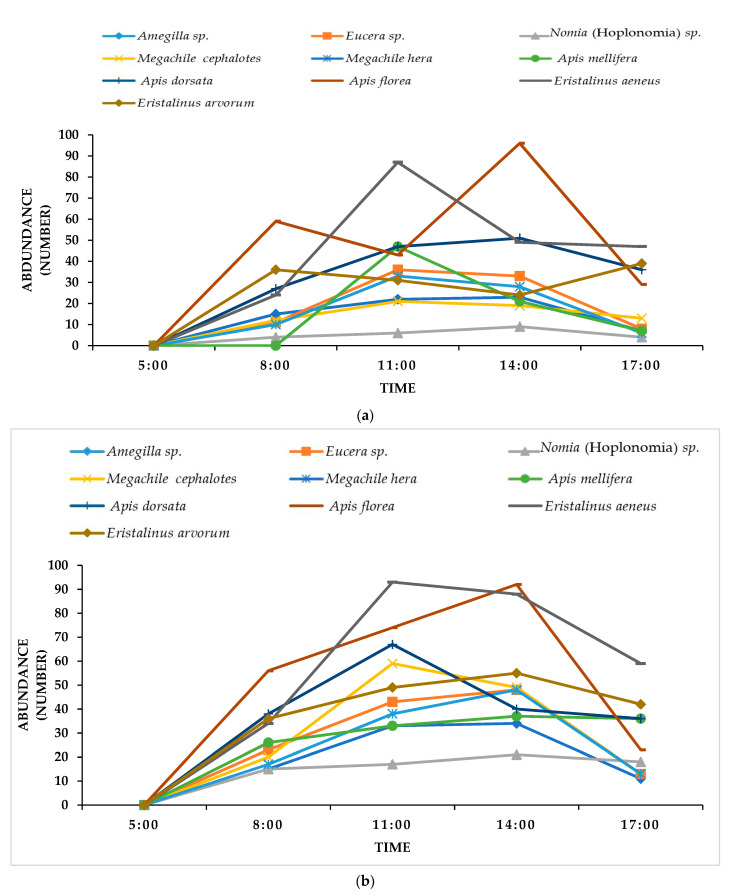
Diurnal dynamic pattern of solitary bees, honeybees and flies in lucerne fields in MNSUAM, Pakistan, during (**a**) 2019, (**b**) 2020.

**Table 1 insects-12-01029-t001:** Insect species in *Medicago sativa* L., with their total abundance and visitation frequency.

Order	Pollinator Group	Family	Genus/Species	Total Abundance (2019)	Total Abundance (2020)	Visitation Frequency (Individuals/m^2^/min) 2019	Visitation Frequency (Individuals/m^2^/min) 2020
Hymenoptera	Solitary bees	Apidae	*Amegilla* sp.	67 ± 3.71	93 ± 5.98	0.022	0.031
			*Eucera* sp.	88 ±7.27	127 ± 8.26	0.029	0.042
		Halictidae	*Nomia* (Hoplonomia) sp.	23 ± 1.18	71 ± 1.25	0.008	0.024
		Megachilidae	*Megachile cephalotes*	65 ± 2.21	141 ± 11.11	0.022	0.047
			*M. hera*	77 ± 6.63	116 ± 8.38	0.026	0.039
	Honeybees	Apidae	*Apis mellifera*	75 ± 10.38	132 ± 2.48	0.025	0.044
			*A. dorsata*	165 ± 5.44	181 ± 7.30	0.060	0.054
			*A. florea*	227 ± 14.45	245 ± 14.72	0.082	0.044
Diptera	Flies	Syrphidae	*Eristalinus aeneus*	207 ± 13.05	274 ± 13.73	0.069	0.091
			*E. arvorum*	130 ± 3.28	182 ± 4.13	0.043	0.061

**Table 2 insects-12-01029-t002:** Comparison of means of ranks of pollination effectiveness in terms of tripping and pollen harvest.

Pollinator Species	Number of Virgin Flowers Tripped/Raceme (*N* = 40)	Number of Already Tripped Flowers Visited/Raceme (*N* = 40)	Number of Pollen Grains Harvested in Single Visit (*N* = 10)
*Amegilla* sp.	31.63 a * (58.51 ± 5.65) **	8.50 f (0.61 ± 0.10)	32.81 c (102.63 ± 4.59)
*Eucera* sp.	32.25 a (62.50 ± 5.46)	8.25 ef (0.75 ± 0.31)	28.38 c (91.63 ± 5.31)
*Nomia* (Hoplonomia) sp.	29.63 a (56.79 ± 3.94)	15.00 def (1.00 ± 0.00)	24.31 cd (79.25 ± 7.12)
*Megachile cephalotes*	31.75 a (59.29 ± 6.31)	8.13 d (1.25 ± 0.13)	72.81 a (1012.88 ± 19.36)
*M. hera*	27.25 a (55.24 ± 3.56)	12.88 de (1.04 ± 0.08)	72.19 a (1007.63 ± 19.76)
*Apis mellifera*	9.75 b (10.12 ± 0.84)	29.50 b (2.25 ± 0.22)	53.38 b (725.50 ± 17.00)
*A. dorsata*	16.13 b (9.05 ± 2.67)	26.25 c (1.75 ± 0.18)	51.75 b (715.38 ± 15.06)
*A. florea*	7.50 b (3.57 ± 0.00)	28.75 c (1.79 ± 0.14)	52.38 b (717.50 ± 12.82)
*Eristalinus aeneus*	11.63 b (1.40 ± 0.68)	30.88 a (3.18 ± 0.28)	6.38 e (16.62 ± 2.00)
*E. arvorum*	7.50 b (3.57 ± 0.00)	36.88 b (2.54 ± 0.15)	10.63 de (21.75 ± 1.75)
Chi-Square	32.69	32.09	74.18
*p*	0.0000	0.0000	0.0000

* Means of ranks in columns having different letters are statistically significant at α = 0.05. ** Mean values with standard errors are given in parenthesis.

**Table 3 insects-12-01029-t003:** Comparison of means of ranks regarding tripping and already tripping per raceme (time wise).

Time	Number of Virgin Flowers Tripped/Raceme	Number of Already Tripped Flowers Visited/Raceme)
8:00	20.20 ab * (33.87 ± 3.73) **	19.15 bc (1.51 ± 0.19)
11:00	22.05 a (35.63 ± 4.45)	24.10 a (1.41 ± 0.12)
14:00	19.85 b (29.00 ± 4.24)	22.2 b (1.77 ± 0.15)
17:00	19.90 b (29.51 ± 3.99)	16.55 c (1.76 ± 0.14)
Chi-Square	0.25	2.44
*p*	0.969	0.490

* Means of ranks in columns having different letters are statistically significant at α = 0.05. ** Mean values with standard errors are given in parenthesis.

**Table 4 insects-12-01029-t004:** Comparison of means of ranks of foraging behavior in terms of visitation rate and stay time.

Pollinator Species	Visitation Rate (No. of Racemes Visited/Min) (*N* = 20)	Stay Time (Time Spent (Seconds) on a Raceme/Visit) (*N* = 40)
*Amegilla* sp.	119.1 ab * (12.40 ± 1.38) **	228.90 abc (10.07 ± 1.06)
*Eucera* sp.	129.3 a (13.75 ± 1.38)	205.89 a (17.66 ± 4.31)
*Nomia* (Hoplonomia) sp.	114.0 ab (12.10 ± 1.34)	248.44 ab (13.70 ± 20.3)
*Megachile cephalotes*	105.61 ab (12.32 ± 1.41)	155.28 abc (9.55 ± 2.20)
*M. hera*	115.43 ab (12.05 ± 1.38)	210.38 abc (10.40 ± 1.59)
*Apis mellifera*	96.53 ab (10.10 ± 1.38)	226.99 abc (9.81 ± 0.98)
*A. dorsata*	68.95 b (7.55 ± 1.38)	213.30 bc (8.41 ± 1.02)
*A. florea*	77.65 ab (8.10 ± 1.38)	213.34 abc (12.02 ± 1.92)
*Eristalinus aeneus*	74.73 ab (7.95 ± 1.41)	119.59 c (4.66 ± 0.72)
*E. arvorum*	103.3 ab (11.05 ± 1.34)	182.91 abc (8.98 ± 1.37)
Chi-Square	22.88	39.39
*p*	0.01	0.00

* Means of ranks in columns having different letters are statistically significant at α = 0.05. ** Mean values with standard errors are given in parenthesis.

**Table 5 insects-12-01029-t005:** Comparison of means of ranks of single visit efficiency of abundant insect pollinator in terms of number of pods per flower, number of seeds set per flower, 1000 seed weight (g) and germination.

Pollinator Species	Number of Pods/Raceme	Number of Seeds/Raceme	1000 Seed Weight (g)	Germination (%)
*Amegilla* sp.	65.88 bc * (3.00 ± 0.27) **	69.81 bc (6.12 ± 0.30)	70.63 a (3.57 ± 0.16)	70.88 a (93.75 ± 2.63)
*Eucera* sp.	64.81 bc (3.00 ± 0.33)	65.50 c (5.87 ± 0.52)	68.06 a (3.50 ± 0.12)	70.88 a (93.75 ± 2.63)
*Nomia* (Hoplonomia) sp.	56.75 cd (2.50 ± 0.19)	56.69 c (4.80 ± 0.35)	64.63 a (3.42 ± 0.17)	56.25 ab (86.25 ± 3.75)
*Megachile cephalotes*	71.88 b (3.50 ± 0.38)	74.00 b (7.63 ± 0.94)	64.75 a (3.45 ± 0.21)	78.88 a (97.50 ± 1.64)
*M. hera*	65.88 bc (3.00 ± 0.2)	69.31 bc (6.12 ± 0.44)	68.00 a (3.53 ± 0.15)	56.25 b (86.25 ± 3.75)
*Apis mellifera*	20.50 fg (0.75 ± 0.31)	19.63 ef (0.87 ± 0.35)	25.56 cd (1.50 ± 0.57)	42.81 ab (77.50 ± 5.26)
*A. dorsata*	36.81 e (1.63 ± 0.26)	35.69 de (2.38 ± 0.60)	49.25 ab (3.08 ± 0.18)	23.19 bc (62.50 ± 5.26)
*A. florea*	19.56 fg (0.75 ± 0.25)	21.25 ef (1.00 ± 0.39)	19.94 cd (1.56 ± 0.48)	31.63 bc (67.50 ± 7.01)
*Eristalinus aeneus*	31.13 ef (1.38 ± 0.18)	28.38 de (1.88 ± 0.69)	33.19 bc (1.92 ± 0.58)	29.63 bc (67.50 ± 5.90)
*E. arvorum*	10.88 g (0.25 ± 0.16)	10.63 f (0.13 ± 0.13)	12.31 d (0.38 ± 0.38)	17.94 c (50.00 ± 9.26)
Open pollination	92.50 a (13.25 ± 0.88)	92.50 a (21.75 ± 1.42)	58.38 ab (3.32 ± 0.11)	58.63 ab (87.50 ± 3.66)
Hand pollination	45.44 de (2.00 ± 0.19)	38.63 d (2.75 ± 0.53)	47.31 ab (3.10 ± 0.18)	45.06 ab (76.25 ± 7.78)
Chi-square	74.22	78.62	49.60	47.08
*p*	<0.0001	<0.0001	<0.0001	<0.0001

* Means of ranks in columns having different letters are statistically significant at α = 0.05. ** Mean values with standard errors are given in parenthesis.

## Data Availability

Not applicable.

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
