# Peer review of "Comparative Efficiency of Native Insect Pollinators in Reproductive Performance of Medicago sativa L. in Pakistan"

_insects, 2021, doi:10.3390/insects12111029_

Round 1
Reviewer 1 Report
Comparative efficiency of native insect pollinators in reproduc- 2 tive performance of Medicago sativa L. in Pakistan
Self-pollination results in low seed set rate in alfalfa. Investigating and breeding high-efficiency pollination insects is crucial to develop alfalfa seed production in local. This paper focused on surveying the native wild insect varieties and the pollination behavior, which is a meaningful work. Some revise suggestions are as follows.
Page 1
Line 12, “cross-pollinating crop” should be revised to “cross-pollinated crop”
Line 15, “pollinator” should be plural.
Line 52-59, “the floral of alfalfa is pollinated unless the keel is forced to open” is suggested to be mentioned in this part.
Line 59, the reason of low efficiency of honeybees for pollinating alfalfa flowers is suggested to be written in the end of the paragraph.
Page 4
Table 1. Apis dorsata and Apis florea should be written in the abbreviation, such as A. dorsata and A. florea. The first appearance with the same genus should be written in full name otherwise in abbreviation.
Page 5-6. Figure 1. The title of vertical axis in (a) and (b) has the different format, with or without bracket.
Line 169. The bracket part in the subtitle is suggested to be deleted.
Page 7-8. Line 190-194 should be moved in front of Table 3.
Line 195. The bracket part in the subtitle is suggested to be deleted. The table 2 is suggested to be separated into two tables, Pollination effectiveness and Foraging behavior.
Author Response
Hi dear
Please find the attachment of improved manuscript as per your good suggestion.
In manuscript your suggestions has been improved with red font color.
Regards
-------------------
Response to Reviewer 1 Comments
(In manuscript these are tackled with font red in colour)
Point 1: Line 12, “cross-pollinating crop” should be revised to “cross-pollinated crop”
Response 1: changed “cross-pollinating crop” to “cross-pollinated crop”
Point 2: Line 15, “pollinator” should be plural.
Response 2: “pollinator” changed to “pollinators”.
Point 3: Line 52-59, “the floral of alfalfa is pollinated unless the keel is forced to open” is suggested to be mentioned in this part.
Response 3: These lines (53-57) added to keep the flow of your precious suggestion.
The lucerne flower has a standard petal on which bees often land, as well as two smaller wing petals on each side. These keel petals exert significant pressure on the female sexual column. The floral of alfalfa is pollinated unless the keel is forced to open i.e., the release of sexual column. The discharge of the sexual column is referred to as 'tripping' the flower.
Point 4: Line 59, the reason of low efficiency of honeybees for pollinating alfalfa flowers is suggested to be written in the end of the paragraph.
Response 4: Reason of low efficiency of honeybees has been inserted in this section as per your suggestions. Line 63-65.
This inefficiency of honeybees in lucerne pollination is because of some adaptive behavioural characters, its side feeding habit for only nectar foraging and lacking of the essential amino acid (isoleucine), in lucerne pollen.
Point 5: Table 1. Apis dorsata and Apis florea should be written in the abbreviation, such as A. dorsata and A. florea. The first appearance with the same genus should be written in full name otherwise in abbreviation.
Response 5: Changes done as per your suggestions. Also needful done for Table 2,4 and 5.
Point 6: Page 5-6. Figure 1. The title of vertical axis in (a) and (b) has the different format, with or without bracket.
Response 6: The title of vertical axis in figure (b) has been improved by inserting brackets.
Point 7: Line 169. The bracket part in the subtitle is suggested to be deleted.
Response 7: needful done
Point 8: Page 7-8. Line 190-194 should be moved in front of Table 3.
Response 8: needful done. Now at line 186-189.
Point 9: Line 195. The bracket part in the subtitle is suggested to be deleted. The table 2 is suggested to be separated into two tables, Pollination effectiveness and Foraging behavior.
Response 9: Line 195. The bracket part in the subtitle has been deleted as per your suggestion.
The table 2 has been separated into two tables with following changes;
Table 2. Comparison of mean values (mean ± SE) of pollination effectiveness in terms of tripping and pollen harvest. (Page 7)
Table 4. Comparison of mean values (mean ± SE) of foraging behaviour in terms of visitation rate and stay time.
(Page 8).

Reviewer 2 Report
Overall a nice contribution. However, there are several issues that should be addressed before to considerer the manuscript worthy of publication. In details:
- Line 99 - please insert 8:00 and 17:00 instead of 800 and 1700. Do the same in lines 110 abd 117 and check all over the text.
- Diurnal abundance of insest pollonator: to me it is not clear how you have determined the specimens. Did you catch the specimens? If so, which was the methodology? Were the specimens preserved in ethanol prior to identification? Please, add details in this section
- Data analysis: did you verify data distribution befor to run ANOVA? If so, how? Did you transform data expressed in % before to run ANOVA? If so, how? Please, add details to improve statistic presentation.
- Line 155-157: you said that you have measured the average visitation frequency as individuals/plant/minute, while in table 1 you have expressed data as individual/m2/minute. Please check and modify. Moreover, please add measure unit into total abundance data (first row) in table 1.
Author Response
Hi dear
Please find the attachment of improved manuscript as per your good suggestion.
In manuscript your suggestions has been improved with light green color.
Regards
------------
Response to Reviewer 2 Comments
(Reviewer 2 comments are in light green font colour)
Point 1: Line 99 - please insert 8:00 and 17:00 instead of 800 and 1700. Do the same in lines 110 abd 117 and check all over the text.
Response 1: Needful done as per your suggestion.
Point 2: Diurnal abundance of insect pollinator: to me it is not clear how you have determined the specimens. Did you catch the specimens? If so, which was the methodology? Were the specimens preserved in ethanol prior to identification? Please, add details in this section.
Response 2: Methodology improved as per your valuable suggestion.
All the bee specimens were first morphotyped and then identified on wings during data recording. All the voucher specimens were preserved and submitted to the Agricultural Museum of the Faculty of Agriculture and Environmental sciences, MNS- University of Agriculture, Multan, Pakistan.
Point 3: Data analysis: did you verify data distribution befor to run ANOVA? If so, how? Did you transform data expressed in % before to run ANOVA? If so, how? Please, add details to improve statistic presentation.
Response 3: The Data Analysis has been re-written with details in the manuscript. For your observations the response is as follows;
Yes, we checked the distribution of our data. The distribution of our data was non normal. However, we applied ANOVA since it is robust to data distribution.
The percentage data (Tripping & Germination) was transformed to square root before applying ANOVA.
Now the manuscript has been improved with transformed data.
Point 4: Line 155-157: you said that you have measured the average visitation frequency as individuals/plant/minute, while in table 1 you have expressed data as individual/m2/minute. Please check and modify. Moreover, please add measure unit into total abundance data (first row) in table 1.
Response 4: Modified unit at line number 15-157 as individuals per square meter per minute.
Here the abundance is in total numbers so I mentioned “Total abundance”. If you suggest, measure unit may be added as “Number” which I have modified in Table 1.

Reviewer 3 Report
The manuscript reports a study of the pollination biology of a common and important plant. The experimental approach is good, and the results make sense. The single-visit study is a useful contribution. I think the tables and figures are good, and appropriate literature is cited. The manuscript needs a moderate amount of editorial work, and I've made a lot of corrections/suggestions on the PDF.
David Inouye

Author Response
Hi dear
Please find the attachment of improved manuscript as per your good suggestion.
In manuscript your suggestions has been improved with light blue color.
Regards
-------------------------
Response to Reviewer 3 Comments
(Track changes done by inserting light blue font colour in manuscript)
Point 1: line 16: Insert “to” at line 16 (that lead seed set of alfalfa”
Response 1: “to” inserted as per suggestion. that lead to seed set of alfalfa at line 15
Point 2: line 18: Write single visit as single-visit
Response 2: needful done at line 17
Point 3: line 20: add as effective pollinators at line end
Response 3: needful done at line 19
Point 4: line 22: insert , after crop
Response 4: , inserted at line 21
Point 5: line 29: add were instead of was
Response 5: “were” written at line 29
Point 6: line 30: Write single visit as single-visit
Response 6: needful done at line 30
Point 7: line 34: Write single visit as single-visit
Response 7: needful done at line 34
Point 8: line 46: add the
Response 8: needful done at line 47
Point 9: line 52 add “the”
Response 9: needful done at line 52-57 as rephrasing the lines “The lucerne flower has a standard petal on which bees often land, as well as two smaller wing petals on each side. These keel petals exert significant pressure on the female sexual column. The floral of alfalfa is pollinated unless the keel is forced to open i.e., the release of sexual column. The discharge of the sexual column is referred to as 'tripping' the flower.
Point 10: line 58 add , after flies
Response 10: needful done at line 61
Point 11: line 69 insert , after reference
Response 11: needful done at line 75
Point 12: line 69: insert study
Response 12: needful done at line 75
Point 13: line 70: Write single visit as single-visit
Response 13: needful done at line 76
Point 14: line 76: Insert This except current
Response 14: needful done at line 83
Point 15: line 76: Insert The
Response 15: needful done at line 83
Point 16: line 77: Insert , before Punjab
Response 16: needful done at line 84
Point 17: line 83: Insert that is except with
Response 17: needful done at line 90
Point 18 and 19: line 88
Response 18: needful done at line 95 as “ Lucerne is widely grown in Pakistan as a fodder crop in different districts of Punjab and Sindh provinces.
Point 20: line 92: add occurs instead occur
Response 20: needful done at line 99
Point 21: line 100. Delete “e” in quadrate
Response 21: needful done by deleting “e” at line 107
Point 22: How did you decide what visits not to observe?
Response 22: There were some species which were present nearby but did not come on our focal plant. Moreover, while recording abundance data many other visitors like butterflies and moths (Lepidoptera), green lacewing (Neuroptera) and some species of beetles (Coleoptera) visited lucerne but we did not entertain these visitors in our data.
Point 23: line 105. Insert “the”
Response 23: needful done at line 115
Point 24: line 106: Insert : the number
Response 24: needful done at line 116
Point 25: line 107: inser , after raceme
Response 25: needful done at line 117
Point 26: line 109 Insert bracket referred to as the tripping
Response 26: needful done at line 119 as “ (referred to as the tripping).
Point 27: line 109: use were instead of was
Response 27: needful done at line 119
Point 28: line 113: You asked before or after a visit occurred
Response 28: needful done at line 123 as “after a single visit”
Point 29: line 119: Write single visit as single-visit
Response 29: needful done at line 129
Point 30: line 120: Write single visit as single-visit
Response 30: needful done at line 130
Point 31: line use land instead of sit
Response 31: needful done at line 134 as “land”
Point 32: Did the visitors visit more than one flower per raceme, or were they chased off after one flower?
Response 32: Yes, occasionally, visitors visited more than one flower per raceme but we maintained only one tripped flower and remove all others.
Point 33: line 126: delete word method
Response 33: needful done at line 136
Point 34: line 127: insert – between open pollinated
Response 34: needful done at line 137 as “open-pollinated”
Point 35: line 130: write petri with capital P
Response 35: needful done at line 140
Point 36: line 134: Inser – between ONE WAY
Response 36: 143-152 (methodology improved)
Point 37: line 135: : Inser – between TWO WAY
Response 37: 143-152 (methodology improved)
Point 38: line 138: insert “a” before LSD
Response 38: 143-152 (methodology improved)
Point 39: line 139: write were instead of was
Response 39: 143-152 (methodology improved)
Point 40: line 140: write capital O for office
Response 40: 143-152 (methodology improved)
Point 41: line 146: write 06 as 6
Response 41: needful done at line 157
Point 42: line 148: write 03 as 3
Response 42: needful done at line 159
Point 43: line 148: write 02 as 2
Response 43: needful done at line 159
Point 44: line 150: write (pithitis) as (Pithitis)
Response 44: needful done at line 162
Point 45: line 163: delete “the”
Response 45: needful done at line 177
Point 46 & 47: write (hoplonomia) in figure 1 as ((Hoplonomia)
Response 46: needful done at page 6 Figure 1 (a) and (b)
Point 48: write (hoplonomia) in Table 2 as ((Hoplonomia)
Response 48: needful done at Table 1, 2 and 4.
Point 49: line 179: Insert ; after time
Response 49: needful done at line 193
Point 50: line 208: delete the word “proved”
Response 50: needful done at line 218
Point 51: write (hoplonomia) in Table 4 as ((Hoplonomia)
Response 51: needful done at Table 5, Page 9
Point 52: line 223: use word “among” instead of between
Response 52: needful done at line 232
Point 53: line 230: insert , after crop
Response 53: needful done at line 238
Point 54: line 230: change reddish to radish
Response 54: needful done at line 238
Point 55: line 237: write (hoplonomia) as ((Hoplonomia)
Response 55: needful done at line 246
Point 56: line 238: insert – between “most abundant”
Response 56: needful done at line 247 as “most-abundant”
Point 57: line 238: use varied instead of varies
Response 57: needful done at line 247
Point 58: line 241-243 Sentence needs to be re-written
Response 58: needful done at line 250-252 as “Social characters of Megachile sp. to collect pollen from the lucerne, its maximum foraging period which coincide with the lucerne blooming also enable it to trip the lucerne flowers more efficiently.
Point 59: line 243-244 (Your goodself commented as “by tripping the flowers”.
Response 59: needful done at line 252-253 as “Since most of the female solitary bees are pollen foragers, they exploit flower by ensuring tripping of Medicago flowers.
Point 60: line 248: insert , after result
Response 60: needful done at line 256
Point 61: line 248: delete “to”
Response 61: needful done at line 256
Point 62: line 251: pollinators may be written as pollinators’
Response 62: needful done at line 259
Point 63: line 259: insert ,
Response 63: needful done at line 268
Point 64: line 268: change returns to return only
Response 64: needful done at line 277
Point 65: line 269: use “when” instead of though
Response 65: needful done at line 278
Point 66: line 271: delete the word “load”
Response 66: needful done at line 278
Point 67: line 273: delete the word :the”
Response 67: needful done at line 281
Point 68: line 284: use “are” instead of is
Response 68: needful done at line 292
Point 69: line 287: Write single visit as single-visit
Response 69: needful done at line 295
Point 70: line 290: Write single visit as single-visit
Response 70: needful done at line 298
Point 71 and 72: line 290: insert , after crop and e.g
Response 71 and 72: needful done at line 298
Point 73: line 293: insert “were”
Response 73: needful done at line 301
Point 74: line 296 which often forage by ‘side-working’
Response 74: needful done at line 304 as “which often forage by ‘side-working’ the flowers.
Point 75: line 305: small o in word “Open”
Response 75: needful done at line 313
Point 76: line 306: insert -
Response 76: needful done at line 313
Point 77: line 307: insert -
Response 77: needful done at line 315
Point 78: line 311: write “to be the”
Response 78: needful done at line 319 As “proved to be the most…”
Point 79: line 307: insert -
Response 79: needful done at line 320
Point 80: line 316: replace “gave” with “developed”
Response 80: needful done at line 324
Point 81: line 368: Delete extra space
Response 81: needful done at line 378-379 By refreshing Mendeley
Point 82: line 369: Italiaze holmesi
Response 82: needful done at line 379
Point 83: line 371: is this subtitle?
Response 83: at line 380, Yes, it is the sub title. Improve by editing.
Point 84: line 473: Sativa written as sativa
Response 84: needful done at line 482
Point 85: line 508: Italiaze genus name
Response 85: needful done at line 519 by italiazing the genus name
Point 86: line 516: duplicated
Response 86: at line 527. No. edited by inserting “and” between them. Now it is as “ Alfalfa and alfalfa improvement”

Round 2
Reviewer 2 Report
Dear Colleagues,
I'm not an expert of statistic, but I feel manuscript still suffering on this issue. I asked to you if you have verified the data distribution before to run ANOVA. You replied that data were not normally distributed. As you know, a postulate of ANOVA analisys is that data have to be normally distributed or, if not, they have to be transformed (if possible) before to run the model. Otherwise, you have to use a dedicated statistic model to data not normally distibuted. Therefore, I ask to you again to insert in the manuscript how you have verified the data distribution and/or how you have transformed data that resulted not normally distributed.
Author Response
Point: Data Analysis
I'm not an expert of statistic, but I feel manuscript still suffering on this issue. I asked to you if you have verified the data distribution before to run ANOVA. You replied that data were not normally distributed. As you know, a postulate of ANOVA analisys is that data have to be normally distributed or, if not, they have to be transformed (if possible) before to run the model. Otherwise, you have to use a dedicated statistic model to data not normally distibuted. Therefore, I ask to you again to insert in the manuscript how you have verified the data distribution and/or how you have transformed data that resulted not normally distributed.
Response: Data Analysis
Data regarding pollination effectiveness of different species in terms of tripping, pollen harvest, foraging behaviour (in terms of visitation rate and stay time) and single visit efficiency of abundant insect pollinator (in terms of number of pods per raceme, number of seeds set per raceme, 1000 seed weight) and germination (%) were initially tested for normality as the basic assumption of ANOVA is that error of the model should be distributed normally. By using QQ-plot and Normality test, we did not find this assumption to be met for our data. Therefore, we used nonparametric methods to test the mean equality. These methods are based upon ranking of the mean. Thus, we used Kruskal-Wallis Test for analysis the data followed by Conover-Iman post-hoc test as corresponding Kruskal-Wallis null hypothesis were rejected. Data were analysed by using computer software XLSTAT (XLSTAT. 2014 version: 5.03).
